# Simultaneous Copy Number Alteration and Single-Nucleotide Variation Analysis in Matched Aqueous Humor and Tumor Samples in Children with Retinoblastoma

**DOI:** 10.3390/ijms24108606

**Published:** 2023-05-11

**Authors:** Michael J. Schmidt, Rishvanth K. Prabakar, Sarah Pike, Venkata Yellapantula, Chen-Ching Peng, Peter Kuhn, James Hicks, Liya Xu, Jesse L. Berry

**Affiliations:** 1Convergent Science Institute in Cancer, Michelson Center for Convergent Bioscience, Dornsife College of Letters, Arts and Sciences, University of Southern California, Los Angeles, CA 90089, USA; mjschmid@usc.edu (M.J.S.); pkuhn@usc.edu (P.K.); jameshic@usc.edu (J.H.); 2The Vision Center, Children’s Hospital Los Angeles, Los Angeles, CA 90027, USA; sbpike@usc.edu (S.P.); ppeng@chla.usc.edu (C.-C.P.); 3USC Roski Eye Institute, Keck School of Medicine, University of Southern California, Los Angeles, CA 90033, USA; 4The Center for Personalized Medicine, Children’s Hospital Los Angeles, Los Angeles, CA 90027, USA

**Keywords:** retinoblastoma, aqueous humor, *RB1* gene, liquid biopsy, somatic copy number alterations, variant calling, targeted sequencing

## Abstract

Retinoblastoma (RB) is a childhood cancer that forms in the developing retina of young children; this tumor cannot be biopsied due to the risk of provoking extraocular tumor spread, which dramatically alters the treatment and survival of the patient. Recently, aqueous humor (AH), the clear fluid in the anterior chamber of the eye, has been developed as an organ-specific liquid biopsy for investigation of in vivo tumor-derived information found in the cell-free DNA (cfDNA) of the biofluid. However, identifying somatic genomic alterations, including both somatic copy number alterations (SCNAs) and single nucleotide variations (SNVs) of the *RB1* gene, typically requires either: (1) two distinct experimental protocols—low-pass whole genome sequencing for SCNAs and targeted sequencing for SNVs—or (2) expensive deep whole genome or exome sequencing. To save time and cost, we applied a one-step targeted sequencing method to identify both SCNAs and *RB1* SNVs in children with RB. High concordance (median = 96.2%) was observed in comparing SCNA calls derived from targeted sequencing to the traditional low-pass whole genome sequencing method. We further applied this method to investigate the degree of concordance of genomic alterations between paired tumor and AH samples from 11 RB eyes. We found 11/11 AH samples (100%) had SCNAs, and 10 of them (90.1%) with recurrent RB-SCNAs, while only nine out of 11 tumor samples (81.8%) had positive RB-SCNA signatures in both low-pass and targeted methods. Eight out of the nine (88.9%) detected SNVs were shared between AH and tumor samples. Ultimately, 11/11 cases have somatic alterations identified, including nine *RB1* SNVs and 10 recurrent RB-SCNAs with four focal *RB1* deletions and one *MYCN* gain. The results presented show the feasibility of utilizing one sequencing approach to obtain SCNA and targeted SNV data to capture a broad genomic scope of RB disease, which may ultimately expedite clinical intervention and be less expensive than other methods.

## 1. Introduction

While RB is considered a canonical cancer, having the first molecularly described tumor suppressor gene (*RB1*), surprisingly, very little is known about the molecular basis underlying the intraocular behavior of this cancer and the varying mechanisms of treatment resistance. This is mainly due to a strict contraindication against invasive biopsy for RB tumors that would risk extraocular cancer spread. Extensive molecular analysis of tumor tissue from advanced enucleated eyes has improved our understanding of RB tumorigenesis. This cancer is predominately driven by biallelic inactivation of the *RB1* gene [1] through either a copy number alteration (CNA) and/or single nucleotide variants (SNVs) that negate RB1 protein function. *RB1* function is crucial to RB pathology, and over 1000 *RB1* SNVs have been identified that contribute to disease pathogenicity [2]. In almost half the cases, the causal event is inherited, greatly increasing the risk of tumor development in both eyes (bilateral disease) [3]. Further, in approximately 2% of patients, RB develops without any *RB1* alterations and is instead driven by oncogenic amplification of *MYCN* on chromosome 2, which is also associated with aggressive tumor behavior [4].

Apart from loss of function alterations to *RB1*, deleterious variants of *BCOR* and *CREBBP* are the two most common co-occurring events at the SNV level [5,6,7]. *BCOR* mutations have been correlated with metastatic potential in RB patients, indicating a more aggressive disease [6], and *CREBBP* plays a role in regulating cell cycle and differentiation [8]. Although more work is needed to understand the impact of these genes in RB pathogenesis, both *BCOR* and *CREBBP* are thought to be tumor suppressors; thus, their inactivation may contribute to disease severity [9].

Despite this body of knowledge of *RB1* mutational contributions to RB, none of it is used to improve the care we provide to these young cancer patients. While genomic analyses at the SNV and CNA levels are routinely used to profile the genetic drivers of other tumors, this is not the current clinical practice for children with RB. In the absence of tumor tissue from a biopsy, identifying these alterations for children with RB was previously not possible. This paradigm changed in 2017 when we, and then others, demonstrated that the aqueous humor (AH), the clear intraocular fluid in front of the eye, is an enriched source of tumor-derived cell-free DNA (cfDNA) in RB eyes that can serve as a liquid biopsy [10,11,12,13,14,15]. We have shown highly concordant somatic CNA (SCNA) profiles with 100 ul AH and matched tumor tissue, and with access to genomic tumor information in eyes undergoing treatment, identified a specific highly recurrent SCNA, chromosome 6p (chr6p) gain, as associated with a 10-fold increased risk of treatment failure requiring surgical removal of the eye [11,16]. This finding underlies the importance of identifying molecular drivers for intraocular disease and highlights the utility of AH in the study of RB disease without enucleation. However, to assay both SCNAs and SNVs in one clinical sample, two separate experimental protocols are required [16], which increases the overall cost and hinders the utility of expedited AH analysis for clinical use.

Methods for systematically profiling genetic disease variants have vastly improved in the next-generation sequencing era. Targeted sequencing has been applied to both AH and tumor samples to profile gene regions of interest (primarily the *RB1* gene) for SNV variants, along with small insertions and deletions that negate RB1 protein function [5,16,17]. With similar intent, SCNA profiling via low-pass whole genome sequencing (WGS) has been utilized to investigate *RB1* loss on chromosome 13, along with additional RB-SCNA signatures (i.e., chr6p gains), in both AH and tumor samples. However, utilizing two distinct methods to inform on SNVs and SCNAs is time-consuming and expensive, which may ultimately delay, or hasten, life-changing treatment decisions for the patient (i.e., enucleation when it is not needed). We sought to simultaneously profile genomic SNVs and SCNAs through a single targeted method to give clinicians actionable information in less time and with less incurred costs.

Here, we present a cohort of 11 patients diagnosed with RB for whom we matched tumor and AH samples. By applying a hybrid capture targeted sequencing approach, we not only systematically profile SNVs along the *RB1* gene, but we also observe high intra-patient concordance of SCNAs between respective AH and tumor samples (Figure 1) and confirm all germline *RB1* variants in each bioanalyte. The targeted technique also enabled the interrogation of *BCOR*, *CREBBP*, and *MYCN*, from which we reported deleterious variants of *BCOR* and *CREBBP* and a focal *MYCN* amplification. Altogether, we show how the combination of targeted SNV and SCNA detection in a single method can be utilized to profile RB and how this genomic information can be used together to understand the disease.

## 2. Results

### 2.1. Patient Cohort Characteristics

This study includes 11 individuals diagnosed with RB at CHLA between 2015 and 2019 (Table 1). Cases were de-identified, and tumor stage and seeding class were reported. All patients in the cohort underwent enucleation with tumor tissue available and had matched AH sampled. Further, patient 33 has two AH samples available: one taken at diagnosis (33-dx) and one at enucleation (33-es) following therapy. Demographic and tumor information is available in Table 1. The median diagnostic age was 22 months (range, 3–35 months). The presence of a germline *RB1* mutation was assayed from routine clinical testing to detect heritable mutations driving each patient’s disease. Three patients harbored germline alterations: cases 9 and 13 harbored inactivating SNVs of the *RB1* gene (p.(M148Vfs*8) and p.(R320*), respectively), while case 1 had a hemizygous deletion of chromosome 13q which resulted in a loss of an *RB1* allele (Table 1). All eyes were enucleated after a median of 95 days (SD = 197.5 days) following therapy.

### 2.2. Copy Number Status Is Concordant in Targeted Sequencing Samples to Low-Pass WGS Samples

From standard low-pass sequencing, which is the gold standard for SCNA determination, all 11 (100%) AH and eight out of 11 (72.7%) tumor samples were identified with one or more SCNAs (Appendix A). The AH of case 28 has a chr20q gain but with no other recurrent RB-SCNA signature (Appendix A). To have confidence in SCNA calls from targeted sequencing reads, we evaluated concordance between SCNAs from targeted reads and SCNAs from low-pass WGS reads of tumor samples (see Section 4). When comparing copy number profiles, we found that the mean values were comparable between methods, with a median concordance of 96.2% (SD = 10.7%; Figure 2A). Further, the median concordance for targeted AH to low-pass WGS AH samples was also high at 97.7% (SD = 3%; *n* = 8; Appendix A); however, only eight samples were analyzed due to differences in sequencing methodologies between the low-pass WGS samples (see Section 4).

Since AH and tumor concordance in low-pass WGS samples have been well documented in the literature [11], we sought to determine the concordance between AH tumor samples using our targeted method. For all sample pairs, we found a median concordance of 89.9% (SD = 23.4%; Figure 2B). Notably, case 11 had low-quality targeted tumor sequencing reads due to limited biological complexity, which led to low AH/tumor concordance. After removing case 11, the median concordance between targeted AH and targeted tumor reads raised to 90.1% (SD = 5.5%).

### 2.3. SCNA Profiles from Targeted Samples Recapitulate Focal Gains and Losses

Targeted sequencing has the potential to miss focal chromosomal alterations due to reduced coverage of the genome. However, we verified the sensitivity of our targeted approach by comparing our SCNA results to those of matched samples generated with low-pass WGS. Using our panel, we were able to capture relevant RB SCNAs in our samples that may drive disease and serve as prognostic indicators. *MYCN* is the primary driver of RB in a small fraction of cases without loss of functional RB protein; however, *MYCN* gains can be both with wildtype RB1 (*RB1*^+/+^/*MYCN^A^*) and as a secondary driver with a loss of the functional RB protein (*RB*^−^/^−^). As with other cancers, a *MYCN* gain serves as a poor prognostic indicator in both settings. Case 48 harbored a focal *MYCN* gain, which was detectable with both low-pass WGS and targeted tumor analyses (Figure 2C). Further, case 50 harbored a focal biallelic loss of the *RB1* gene on chromosome 13 that was detected in both low-pass WGS and our targeted approach (Figure 2D).

### 2.4. Inactivating SNVs to the RB1 Gene

While patients harbored many shared passenger mutations between their respective AH and tumor samples, we were interested in detecting disease-driving variants for RB. We found that six out of the 11 cases (54.5%) harbored at least one inactivating *RB1* mutation (Figure 3). In total, we detected nine pathogenic variants, of which eight were shared between the AH and tumor (88.9%; Figure 3). One deleterious variant was detected in the tumor from case 28 p.(R255*) but not in the corresponding AH. Visualization of sequencing coverage via IGV confirmed the SNV is present in the tumor but not the AH (Appendix A).

In general, when the patient harbored a shared AH and tumor mutation, the variant allele frequency (VAF) for the mutation was higher in the AH (Figure 3A). Interestingly, we were able to detect a germline frameshift mutation for case 13 p.(M148Vfs*8) in both the AH and tumor, but the VAF for the tumor was 6.7%, which was significantly lower than 56.5% observed in AH, suggesting a higher normal (i.e., non-RB) cell fraction existing in the tumor sample (Figure 3A). For case 9, we detected a germline mutation p.(R320*) and a second deleterious mutant p.(R251*); the VAF was higher in the AH for both mutants.

Case 33 was a unique patient because we had an AH sample from diagnosis (case 33-dx) and one from secondary enucleation included in the analysis (case 33-es). A splicing variant (c.2325 + 1G>A) was identified in all samples, with high read depths found in the tumor sample and the AH sample at diagnosis (33-dx; Figure 3A,B). Although the targeted sample for case 33-es failed QC (7 total reads were mapped to the genomic location, which was below our cut-off of 10; Figure 3B), we retained this variant since it was present at diagnosis (case 33-dx) and in the enucleated tumor tissue with high confidence. Similarly, we detected a deleterious variant p.(R455*) on the *RB1* gene for case 15 that was only present in the AH but not the tumor (VAF = 90%) that also failed QC (nine reads mapped). Since this variant had low read depth and was only present in the AH and not the tumor, it was discarded; further analysis for case 15 may prove whether this mutation is legitimate.

### 2.5. RB-SCNA Signatures Observed in the Cohort

Genomic and genetic alterations for these RB cases are summarized in Figure 4, showing RB signatures, including RB-SCNAs and/or *RB1* SNVs that were identified in 10 (90.1%) AH samples and nine tumor samples (81.8%). While our methods detected a *MYCN* gain for case 48, immunohistochemical staining of the tumor tissue for RB protein showed a lack of nuclear staining. Thus, while an SNV was not detected, there was a loss of functional RB protein suggesting the *RB1* gene may have been epigenetically silenced and not that this is a tumor driven primarily by *MYCN* [18]. Finally, although the AH of case 28 shows neither RB-SCNA nor *RB1* SNVs, it does have a significant gain on chromosome 20, indicating the existence of somatic alteration (Appendix A).

### 2.6. Deleterious Variants Detected That Impact BCOR and CREBBP

In addition to *RB1* variants, we detected deleterious SNVs (*n* = 3) and SCNAs (*n* = 2) to the *BCOR* or *CREBBP* genes in 5 cases (45.4%). All SNV variants were shared between AH and tumor samples (Figure 5A,B). We also detected a chromosome 16p loss in case 50 (Figure 2D) and case 8 (Figure 5C) that resulted in the loss of a *CREBBP* allele. No activating *MYCN* SNVs were detected in the cohort.

## 3. Discussion

We report the multi-sample genomic analysis of RB-related disease drivers at SCNA and SNV levels by applying a single targeted sequencing method and provide further evidence that the AH can be used as a liquid biopsy approach for the non-invasive evaluation and monitoring of RB disease. This study is the first to examine the SCNA status of both AH and tumor samples while simultaneously analyzing *RB1*, *BCOR*, *CREBBP*, and *MYCN* for SNV disease drivers. We were able to confidently call RB-signature variants for 10/11 patients in the cohort, including inactivating *RB1* gene variants in nine cases and a focal *MYCN* gain for case 48. Case 15 had no detectable *RB1* variants. There are multiple reasons why pathogenic variants are not detected in all cases, including known epigenetic-related silencing events that inactivate RB1 protein function [18].

SCNA profiling of both AH and blood cfDNA has been previously established as an effective liquid biopsy approach by our group and others. In the present study, we found that our target capture approach was highly concordant with the current gold standard of low-pass WGS for calling SCNAs in AH and tumor samples from RB patients (Figure 2A). We were also able to confirm all germline *RB1* CNAs that were detected by the clinical blood test for both the AH and tumor in our assay (Table 1, Figure 3 and Figure 4). Finally, the targeted approach was able to detect a focal *MYCN* gain in case 48 (Figure 2C), highlighting the sensitivity of the assay.

SNV profiling showed strong concordance in the detection of pathogenic *RB1* variants between the AH and tumor samples (Figure 3). Taken together, we found eight out of nine (88.9%) disease-driving SNVs in the AH (88.9%), thereby further supporting AH as an alternative liquid biopsy approach for RB. Case 28 was the one outlier where we found a deleterious variant p.(R255*) that was detected in the tumor with a low fraction but not AH (Figure 3 and Appendix A). However, AH SCNA, in this case, harbors clear evidence of the somatic alteration of the disease.

Overall, the VAFs in AH samples were found to be higher than those in matched tumor samples. This observation could be attributed to either the presence of normal cells in the tumor biopsy, which could lower the VAFs, or poor sequencing quality leading to noisy data. For instance, in case 49, the copy number profiles show that the ratio of the median of the segmented values in the tumor is much less than that of the AH sample (Appendix A: Case 49 tumor versus AH). The tumor profile appears to be condensed compared to the AH profile. At the SNV level, the tumor VAF for the p.(E440*) mutation in case 49 was only 12.7%, whereas the AH VAF was much higher at 43.1% (Figure 3). These findings further support the possibility of non-RB cells, such as associated retinal cells contaminating the tumor sample during preparation; this will be especially likely in secondarily enucleated eyes wherein the majority of the tumor has undergone necrosis, and the specimen is highly calcified. We also identified a germline SNV in case 13 (p.(M148Vfs*8)) that was confirmed by our analysis (Table 1). However, the VAF of this mutation was found to be significantly lower in the tumor sample (6.7%) than in the AH sample (56.5%; Figure 3). We speculate that this discrepancy could be attributed to the presence of more non-cancerous cells in the tumor sample, as evidenced by the respective copy number profiles of the tumor and AH samples (Appendix A: Case 13 tumor versus Case 13 AH). Specifically, the tumor samples appeared nearly diploid, while the AH samples displayed clear RB-SCNA signatures.

While the blood has been utilized to profile *RB1* somatic SNVs [19], the data presented here suggests the AH is superior due to its high tumor fraction. Even with the high-quality data observed in the AH, a substantial amount of signal from the disease-driving splicing variant from case 33-es (c.2325 + 1G>A) was lost following treatment, which underlines the challenge and needs for high DNA concentrations for mutational analysis [20]. Though direct AH-blood SCNA comparisons have been done previously [21], further direct comparisons between the blood, AH, and tumor in SNV analysis for RB are needed.

While this method provided promising results, improvements can be made that can increase its accuracy. For example, probe optimization would benefit future studies. An alternative probe design, which encompasses boundaries slightly outside the genes of interest, could increase efficiency. Extending probe coverage would also allow for the profiling of potential regulatory sequences in the intronic regions of the genes. Lastly, deep sequencing in an unbiased approach, such as whole exome or whole genome sequencing, would be more expensive but may uncover disease-driving mutants outside the scope of the four genes we profiled in this study.

Retinoblastoma tumor tissue cannot be biopsied due to the risk of metastatic spread, which means that obtaining tumor tissue for molecular analysis requires surgical removal of the eye. However, the primary objective of therapy is to treat cancer while preserving the eye, which precludes access to tumor tissue. Our analysis further supports the realization that AH samples collected in vivo can serve as a liquid biopsy for RB, and we have demonstrated the feasibility and accuracy of a combined SCNA-SNV analysis from a single sample and single targeted assay in a time-efficient and cost-effective manner [22,23]. These findings represent a significant clinical advance in the diagnosis and treatment of RB, with the potential to improve outcomes and quality of life for patients and their families.

## 4. Materials and Methods

Statement of research ethics: The established biorepository and collection of coded clinical data was approved by the Children’s Hospital Los Angeles Institutional Review Board. This study adhered to the tenets of the Declaration of Helsinki and was in accordance with the Health Insurance Portability and Accountability Act. All patients provided written informed consent via a legal guardian for all procedures performed.

Sample collection and processing: Biospecimens were collected as previously reported [16]. Following specimen extraction, samples were stored at −80 °C until processing. All samples underwent cfDNA isolation and sequencing within 1 month of extraction. cfDNA extraction and processing was previously described [16].

Probe design and efficiency: Targeted SNV probes were designed by Agilent SureSelect DNA (Agilent, Santa Clara, CA, USA) [24] and covered portions of the *RB1*, *MYCN*, *BCOR*, and *CREBBP* genes in accordance with the human genome reference version hg19. Probes summed to cover 55,097 bp in total (Appendix A).

Copy number alteration analysis: Copy number variation was performed on the targeted reads with the R package CopywriteR (version 3.16)—which takes advantage of off-target reads in targeted sequencing data—using the suggested parameters [25]. Samples were aligned with BWA-MEM (version 0.7.17). Bins were set to 500 kb for the hg19 genome, and a female gDNA control with no copy number alterations was used as a baseline for all samples (Appendix A).

WGS reads were analyzed as previously described [26] with some modifications. Briefly, samples were 150 base-pairs paired-end sequenced at a depth of 1–2 million reads on Illumina HiSeq 4000. Sequencing reads were aligned via BWA-MEM to the hg19 reference. Reads were mapped to 5000 bins spanning the human genome and were then normalized for GC content. Count data were segmented via the R package DNAcopy (version 1.70.0) [27].

SCNA concordance determination: We measured concordance between copy number profiles as previously described [11]. For comparing the targeted versus low-pass sequencing data, we divided the segmented reads from the targeted sample by the WGS sample. Analogously, for comparing tumors to AH samples, segmented means were divided for the tumor by the segmented means for the respective AH. Samples in question were considered concordant if their ratio was between 0.8–1.2 within each bin. Concordant bins were then set to 1; bins whose ratios were either below 0.8 or above 1.2 were considered discordant and set to 0. Each bin was then normalized for size and then summed together to give a final concordance value between 0–1.

RB-SCNA signatures—defined here as 1q gain, 2p gain, 6p gain, 13q loss, and/or 16q loss—were manually inspected for concordance between tumor and AH samples in both the shallow and targeted sequencing approaches. Other RB-SCNA (i.e., 17q gain and 7q gain) signatures may exist [15], but we did not evaluate the signature concordance between samples for those alterations.

Concordance was not calculated for three pairs of AH-targeted and low-pass WGS samples from cases 11, 49, and 50. These samples were single-end 50 bp sequenced; therefore, concordance metrics would not reflect the rest of the cohort that was paired-end 150 bp sequenced (Appendix A).

Single nucleotide variant calling: Targeted samples were paired-end sequenced at 150 bp on an Illumina HiSeq 4000 (Fulgent, Inc., Temple City, CA, USA). FastQ (version 2.0.1) files were aligned to hg19 with BWA-MEM (version 0.7.17). Reads falling within the probed regions (*RB1*, *BCOR*, *CREBBP*, and *MYCN*) were considered. SAMtools (version 1.15-1) was used to sort by coordinates [28]. Picard (version 2.27.4) was utilized to mark duplicates. The bam files were then indexed via SAMtools and utilized as the input for the Mutect2 (version 4-4.3) pipeline. For Mutect2 analysis, both the germline resource and panel of normals provided by GATK were utilized [29]. The read orientation (via the learn orientation model), pile-up summaries, and contamination artifacts were all calculated and utilized for the filtering of Mutect2 variant calls. The variant effect was determined through the Annovar pipeline (version 2020-06-08) [30], and specifically, SIFT, Polyphen2, and gnomAD scores were utilized to classify mutation severity (i.e., deleterious or tolerable) [31,32,33]. Unless otherwise specified, mutations of interest were then filtered to require a read depth greater than 10. Final mutation calls were then viewed through the Integrated Genome Viewer to confirm the calls [34]. For further validation, all Mutect2 calls were cross-checked with an internal bioinformatic pipeline at the CHLA Center for Personalized Medicine [35].

## Figures and Tables

**Figure 1 ijms-24-08606-f001:**
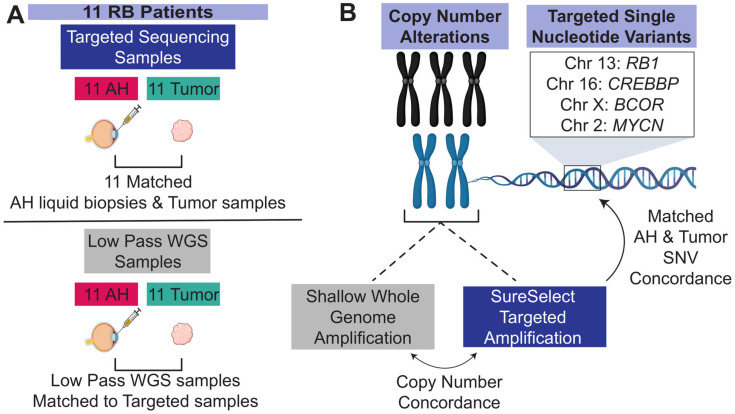
Schematic of study design. (**A**) Eleven patients were present in the cohort. ((**A**) **Top**) Sample counts for each biospecimen collected that was processed through the dual-targeted SNV and SCNA methods are displayed. Eleven patients contained matched AH and tumor-targeted sequencing samples. ((**A**) **Bottom**) Sample count of low-pass WGS sequencing samples. (**B**) SCNAs from the SureSelect Targeted Sequencing approach were compared to the gold standard low-pass WGS, and concordance was calculated. The targeted reads were then used to call SNVs, and the 11 matched AH and tumor-targeted reads were directly compared for concordance. This figure was generated with BioRender.

**Figure 2 ijms-24-08606-f002:**
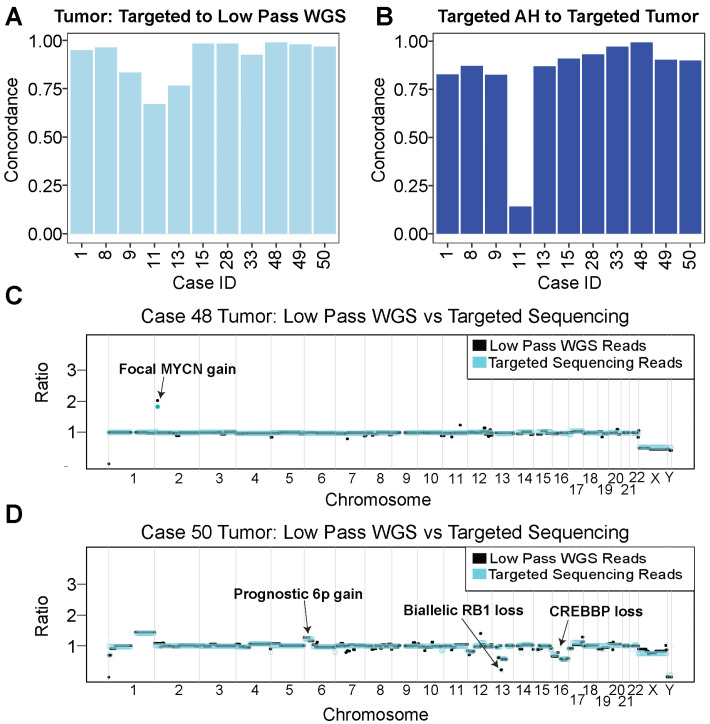
Copy number alteration analysis shows high concordance between targeted AH and tumor samples. (**A**) Concordance of derived tumor SCNAs between targeted sequencing reads and their matched low-pass WGS reads. (**B**) Concordance of SCNAs between targeted AH sequencing samples and their respective targeted tumor sequencing samples. (**C**) Copy number representation for case 48’s tumor sample overlaying targeted sequencing reads (teal) to low-pass WGS reads (black). A focal *MYCN* gain was detected on chromosome 2 with both methods. (**D**) Copy number plot for case 50’s tumor sample overlaying targeted sequencing reads (teal) to low-pass WGS reads (black) showing focal *RB1* gene deletion on chromosome 13 and *CREBBP* loss on chromosome 16.

**Figure 3 ijms-24-08606-f003:**
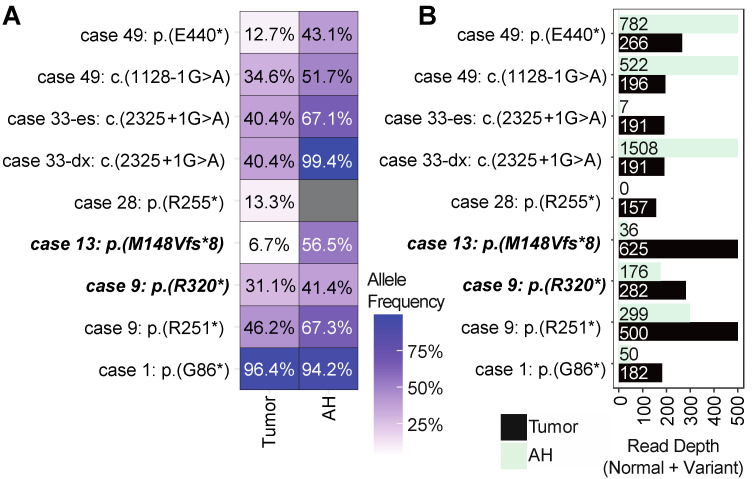
SNV landscape of RB patients: (**A**) Allele frequency plot for deleterious SNVs detected to the *RB1* gene. Case 33 had two AH samples (pt33-dx and pt33-es) but had a single tumor sample that was taken at enucleation. Cases that are ***bold*** and *italicized* are ***germline variants***. Cases with asterisk (*) indicate nonsense variants that result in truncated protein. Dark grey boxes indicate the variant was not found in that sample. (**B**) Read depth for each genomic site in (**A**). Depth accounts for total reads at each specified site, including both variant reads and normal allele reads. X-axis for depth was cut off at 500 bases for legibility.

**Figure 4 ijms-24-08606-f004:**
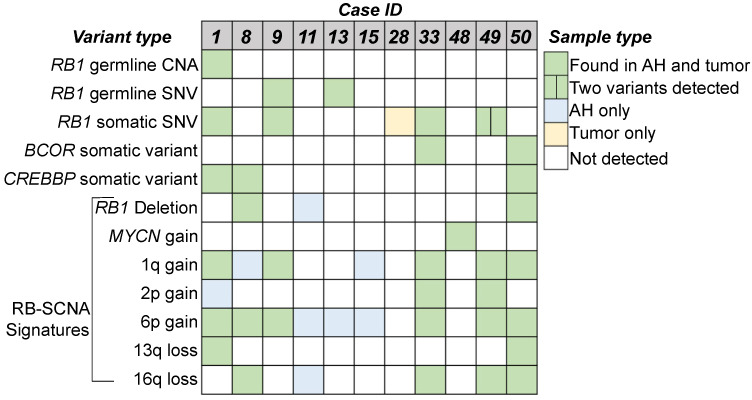
Summary of variants observed in the Cohort. Germline and somatic *RB1* SCNAs and SNVs, and other RB-signature genetic variants are displayed. Germline variants detected in blood (separate clinical diagnostic test), AH, and tumor for cases 1, 9, and 13. Cases 33-dx and 33-es were merged; they have the same *RB1* somatic SNV detected at diagnosis and enucleation. Variants depicted are found in both targeted and tumor samples.

**Figure 5 ijms-24-08606-f005:**
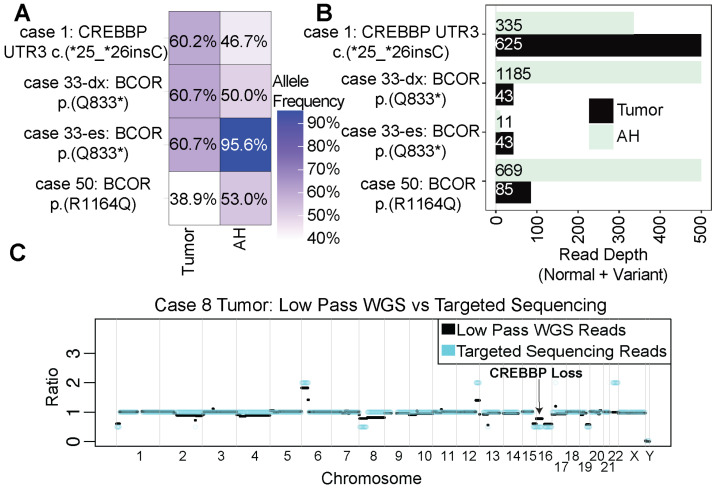
Mutational landscape to *CREBBP* and *BCOR*: (**A**) Allele frequency plot for deleterious mutations to *BCOR* and *CREBBP* detected in the cohort. Cases with asterisk (*) indicate nonsense variants that result in truncated protein. (**B**) Read depth for each genomic site in (**A**). Depth accounts for total reads at each specified site, including both variant reads and normal allele reads. X-axis for depth was cut off at 500 bases for legibility. (**C**) SCNA plot for case 8 displaying a loss to chromosome 16 impacting *CREBBP*.

**Table 1 ijms-24-08606-t001:** Clinical characteristics of the 11 patients in the cohort.

Case ID	Sex	Age (Months)	Laterality ^1^	IIRC Group	TNM Stage	Seed Class	Enucleation	Laser ^2^ Sessions	Cryo ^2^ Sessions	IVM ^3^	Germline *RB1* Blood Test
1	M	20	U	E	CT3C	None	Primary	-	-	No	13q-
8	F	22	U	D	CT2B	Dust	Secondary	10	0	Yes	Negative
9	F	29	B	D	CT2B	Dust	Secondary	2	2	No	p.(R320*)
11	F	8	U	D	CT2B	None	Secondary	15	4	Yes	Negative
13	M	34	U	D	CT2B	Sphere	Secondary	8	1	Yes	p.(M148Vfs*8)
15	M	10	U	D	CT2B	None	Secondary	16	1	Yes	Negative
28	F	3	B	D	CT2B	None	Secondary	7	1	Yes	Negative
33 ^4^	M	22	U	D	CT2B	Sphere	Secondary	3	1	Yes	Negative
48	M	18	U	D	CT2B	All; cloud	Secondary ^5^	-	-	No	Negative
49	M	35	U	D	CT1B	All; cloud	Primary	-	-	No	Negative
50	F	24	U	D	CT2B	None	Primary	-	-	No	Negative

^1^ Laterality: Unilateral (U) and bilateral (B). ^2^ Number of laser and cryotherapy sessions received before eye enucleation. ^3^ IVM: Intravitreal Melphalan injections (chemotherapy) were received (yes) or not received (no). ^4^ Patient 33 has two AH samples: at diagnosis (33-dx) and at enucleation (33-es). ^5^ Patient 48 received one cycle of intra-arterial chemotherapy and then was secondarily enucleated.

## Data Availability

The data presented in this text are available in the text and Appendix A. Additional data and analysis pipelines and scripts utilized in this manuscript are available upon request.

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
