# Peer review of "Simultaneous Copy Number Alteration and Single-Nucleotide Variation Analysis in Matched Aqueous Humor and Tumor Samples in Children with Retinoblastoma"

_ijms, 2023, doi:10.3390/ijms24108606_

Round 1

Reviewer 1 Report

The authors present a one-step targeted sequencing method for identifying somatic copy number alterations (SCNAs) and SNVs in retinoblastoma-relevant genes in cfDNA from aqueous humor (AH) and DNA from tumors. They analyzed paired tumor and AH samples from 11 RB eyes. They show that, concerning SCNAs, results obtained by the new method are concordant with those of the current standard, low-pass whole genome sequencing. They show that SNVs are shared between AH and tumor samples and that RB-SCNA signatures show high concordance. The one-step targeted sequencing method thus is an improvement over current techniques.

Some suggestions and minor points:

1.) Fig 2C: Patient 48 Tumor: Low Pass WGS vs. Targeted Sequencing: the figure states „Focal MYCN gain“ but the legend states „focal MYCN amplification“. IMO,at a ratio of 2, gain is more correct. (Also see line 175)

2.) Fig 3B: Correct that read depth is the number of reads with the variant sequence? In some AH, read depth is very low, and the light coloration makes it difficult to detect the small bars. I would consider adding the number of reads as text.

3.) Fig 3A: Correct that in case 33-es, the SNV (c.2325+1G>A) is detected in AH with a VAF of 67.1% and with 7 reads that show the variant sequence? I would reconsider providing point estimates of VAF with a precision (decimals shown) that fits the number of observations underlying these estimates.

4.) Fig 3A: In case 13, the germ-line variant p.M148Vfs *8 is detected in the tumor at a VAF of just 6.7%. This observation merits to be discussed. (analogous situation in tumor of case 9: p.R320*).

5.) Further to the suggestions above (4.) and noteworthy: it appears that results from tumor samples obtained after secondary enucleation pose problems (eg, VAF does not meet theoretical expectations). Also, the tumor of case 49 with primary enucleation (p.E440* at VAF 12.7%). 

6.) Some topics in the discussion appear to be somewhat repeated, e.g., line 239ff and line 248ff.

7.) Is it possible to point out some specific use cases for this test? Just an example to explain how this approach provides „vital genomic information for reliable clinical decision making“ (line 296)?

8.) In the ijms-2370286-supplementary: „Case 1 tumor: Low pass WGS“ and „Case 1 AH: Low pass WGS“ appear to have some similar aberrations (e.g., germline RB1 deletion), but there appear to be striking differences (e.g., chr 6 and chr 16). Why?

Reviewer 2 Report

A well written paper describing a single step targeted sequencing method to detect both SCNAs and SNVs in RB.  Comparison of matched AH and tumour samples is also very useful confirming once more that AH seems to be suitable for routine utilisation in RB management.

Minor points/queries:

1-      Line 56 Ref(3) comes before Ref(2) in line 58.

2-      Table 1 ‘IVM’ needs defining.

3-       Line 208 ‘R’ is missing from p.(R455*).

4-      Table 1 case 50 germline RB1 blood test negative.  Fig 4 however shows case 50 as having a RB1 germline CNA.

5-      Case 48 – lines 174-176 – resulted from focal MYCN amp.  Table 1 gives age as 18m.  This seems late onset for a MYCN amp tumour?

6-      The nomenclature for the putative consequence needs correcting in Table 1 and throughout the paper. There should be brackets after p. eg p.(R320*).

7-      The use of ‘stop-gain’ term could be confusing as ‘gain’ is used in the context of ‘gain on chromosome 20’ and ‘gain on function’ etc.
